# Combating Vaccine Hesitancy with Vaccine-Preventable Disease Familiarization: An Interview and Curriculum Intervention for College Students

**DOI:** 10.3390/vaccines7020039

**Published:** 2019-05-12

**Authors:** Deborah K. Johnson, Emily J. Mello, Trent D. Walker, Spencer J. Hood, Jamie L. Jensen, Brian D. Poole

**Affiliations:** 1Department of Microbiology and Molecular Biology, Brigham Young University, Provo, UT 84602, USA; deborahkj@gmail.com (D.K.J.); emilyjmello@gmail.com (E.J.M.); tdwalker11@charter.net (T.D.W.); spencer.hood@box.hood.org (S.J.H.); 2Department of Biology, Brigham Young University, Provo, UT 84602, USA; jamie.jensen@byu.edu

**Keywords:** vaccine hesitancy, college student, vaccine, vaccine curriculum, interview intervention

## Abstract

In 2019, the World Health Organization (WHO) listed vaccine hesitancy in its top ten threats to global health. Vaccine hesitancy is a “delay in acceptance or refusal to vaccinate despite availability of vaccination services”. Urban areas with large amounts of vaccine hesitancy are at risk for the resurgence of vaccine-preventable diseases (VPDs). Many vaccine-hesitant (VH) parents may be unfamiliar with the consequences of VPDs, and thus might be swayed when confronted with the symptoms and dangers of VPDs. As such, we sought to educate college students (future parents) in an urban vaccine-hesitant hotspot by assigning them to interview family or community members who had experienced a VPD. Student vaccine attitudes were assessed by surveys before and after the interviews. Vaccine-hesitant students who conducted a VPD interview but received no additional vaccine educational materials were significantly more likely (interaction term *p* < 0.001) to become pro-vaccine (PV) (68%) than students who conducted an autoimmune interview and received no additional educational materials. Additionally, students whose interviewees experienced intense physical suffering or physical limitations or students who were enrolled in a course with intensive VPD and vaccine curriculum had significantly increased vaccine attitudes. This suggests that introducing students to VPDs can decrease vaccine hesitancy.

## 1. Introduction

Vaccines are victims of their own success. Due to the effectiveness of vaccination programs, many people have limited or no experience with vaccine-preventable diseases (VPDs) [1]. Parents increasingly assume that the risks associated with VPDs are minimal compared to potential health and safety risks of vaccinations themselves [2,3,4,5]. This has led to a rise in vaccine hesitancy by parents that results in a “delay in acceptance or refusal to vaccinate despite availability of vaccination services” [6]. Urban centers with large clusters of vaccine-hesitant individuals are particularly vulnerable to VPD outbreaks among exposed, unimmunized children. In the 2016–2017 school year, Utah County (Provo) in Utah, USA ranked sixth nationally for the total number of entering kindergartners that were under-vaccinated as measured by non-medical exemption (NME) waivers (*n* = 662 NME) [7]. As many of these parents may have never experienced VPDs, we hypothesized that designing an intervention for college students (future parents) in Provo, Utah might help improve vaccine attitudes and future vaccine uptake for themselves and their families.

Influencing students before they become parents will likely encourage pro-vaccination behaviors for their future and current families, as children and adolescents who participate in health education activities in school can positively influence family health management [8,9,10]. However, there is evidence that correcting erroneous assumptions about potential health and safety risks may be ineffective, causing a “backfire effect” and further entrenching vaccine-hesitant individuals’ beliefs [11,12]. Vaccine hesitancy often arises from deep-rooted ideological beliefs and conspiracist ideational tendencies. As this kind of thinking has been shown to become further entrenched when those holding it are presented with contradictory information, correcting misinformation is often counterproductive [13,14,15]. Further, it is difficult to provide convincing data for the absence of risk; consequently, vaccine-hesitant parents may be recalcitrant to messages aimed at alleviating concerns about vaccine safety and side effects [16,17]. Rather, pro-vaccine interventions may be more effective if they warn of health dangers to individuals from VPDs [17]. Additionally, the vaccine-hesitant movement uses emotionally charged stories with dire long-term consequences to effectively convey anti-vaccine ideology. Combating this rhetoric with a similarly emotional appeal may be an effective preventative strategy [18]. Therefore, we predicted that hearing about the effects of VPDs from family and community members who suffered from VPDs would improve the students’ attitudes towards vaccination. We further predicted that classroom education could improve attitudes towards vaccines.

In this study, we analyzed three courses with different vaccine instructional approaches varying from none to intensive. Enrolled students were also assigned to interview a family or community member about their personal experience with either an autoimmune disease (control) or a VPD. Students exposed to either intensive VPD-focused vaccine instruction or who interviewed individuals who had had a VPD had statistically significant and meaningful gains in vaccine attitude.

## 2. Methods

### 2.1. Study Population

Students at a large private institution in the Western United States were enrolled in a quasi-experimental survey-based intervention study. Students were eligible if they attended one of three courses in the Winter 2018 semester: General education non-science major Bio 100 (Principles of Biology, one section), and microbiology and molecular biology (MMBio) major-specific courses MMBio 240 (Molecular Biology, two sections) and MMBio 261 (Infection and Immunity, one section). Vaccination principles were taught in Bio 100 and MMBio 261, but not in MMBio 240. All students enrolled in the courses were encouraged to participate and offered extra credit for their time and efforts. Our study sample consisted of 425 students who completed the study requirements (574 began the study). The study was conducted in accordance with the Declaration of Helsinki, and study procedures were approved by the Institutional Review Board at the institution (study #E17263). All participants received and signed a consent form that included a description of the study and were assigned a random number to protect their identities. Students were informed that their instructors would never see their names associated with any of the survey results, and steps were taken to avoid such instructor access.

### 2.2. Assigning Vaccine Attitude Groups and Randomization Process

To determine initial vaccine attitudes, students took a pre-interview survey (see Box 1) and were asked to rate each question from 1–5 where 1 is “strongly disagree” and 5 is “strongly agree”. Each question concerning vaccine attitude was chosen to cover a specific aspect of vaccine hesitancy. Pre-intervention vaccine attitude scores (VASs) were tallied from questions 1, 4, 9, 11, and 13 (Box 1). Question 1 is a test of general attitude towards vaccines. Question 4 addresses side effects, relevant since many vaccine-hesitant individuals are afraid of these side effects. Question 9 is about the common belief that vaccines cause autism, a major concern for many vaccine-hesitant people. Question 11 gives an opportunity for the participants to opine on the positive aspects of vaccination in terms of how well they work. Question 13 is a question about how the vaccine attitude would affect action and give it a more real-world, rather than theoretical, effect. To avoid answer bias, students were not informed that the study was about vaccination opinions and additional questions about autoimmune diseases and depression were included in the survey. Scores from questions 4 and 9 were reverse coded to account for the negative nature of the question. These questions were written in a negative way to avoid biasing the study by presenting vaccines in only a positive light in the questions. Students with VASs between 20 and 25 points were categorized as “Pro-Vaccine” (PV) and students that had a VAS less than 20 points were categorized as “Vaccine-Hesitant” (VH). A cutoff value of 20 was chosen because it meant, on average, that the student at least “agreed” (score or converted score of 4) with all of the vaccine attitude questions. A score of less than 20 would mean that the student on average either did not at least “agree” with the pro-vaccine statements, or that they had a serious disagreement with at least one statement about vaccines. Students were assigned to interview groups by alternating autoimmune (negative control for survey and interview effects) and vaccine-preventable disease (VPD) interview assignments alphabetically within PV and VH categories so that equal numbers of students were assigned to each interview intervention.

Box 1Pre-interview survey.Rate each question from 1–5 where 1 is “strongly disagree” and 5 is “strongly agree”Vaccines are more helpful than harmfulTreatment for autoimmune diseases is more helpful than harmfulMedications for depression are more helpful than harmfulVaccines often have severe side effectsPeople with autoimmune diseases suffer considerablyMedication for depression is effective at treating depressionThere are effective treatments for autoimmune diseasesDepression can be overcome using willpowerVaccines cause autismExercise is the best treatment for autoimmune diseasesVaccines are effective at preventing diseaseMedications for depression have severe side effectsI am likely to fully vaccinate my children/I have fully vaccinated my children

Students were emailed their survey assignments and related paperwork, and based on their group assignment, were asked to interview members of the community who had experienced either a VPD or an autoimmune disease before the end of the semester using the interview questions shown in Box 2. To encourage study completion, students received full points for extra credit with completed survey submission (Bio 100 10 points, 1% of grade; MMBio 240 20 points, 2.3% of grade; MMBio 261 20 points, 2.3% of grade). At the end of the semester, students were administered a post-interview survey (see Box 3) that reiterated the pre-interview survey questions and included follow-up questions about the survey itself. These questions (14–18 in the post-interview survey) were written to identify the aspects of the interview that had the most significant impact on vaccine attitudes. They provided both an opportunity to rank several factors and the ability to explain in their own words how the various aspects of the interview affected them. We then assigned students a post-intervention VAS and assessed for changes in overall vaccine scores between the pre-and post-intervention surveys. Included in this analysis was determining whether students moved from the vaccine-hesitant to pro-vaccine group, or vice-versa.

Box 2Interview questions.What is your relationship to the person who had the disease?When did they develop the disease?Which disease was involved?What type of physical suffering did the disease cause? How bad was it?How did the disease limit the person’s ability to do normal activities?How did the disease affect the person’s interaction with other people?How did the disease affect the person’s friends, family, or loved ones?How did the disease affect the person financially?Were there any other effects of the disease?

Box 3Post-interview survey.Vaccines are more helpful than harmfulTreatment for autoimmune diseases is more helpful than harmfulMedications for depression are more helpful than harmfulVaccines often have severe side effectsPeople with autoimmune diseases suffer considerablyMedication for depression is effective at treating depressionThere are effective treatments for autoimmune diseasesDepression can be overcome using willpowerVaccines cause autismExercise is the best treatment for autoimmune diseasesVaccines are effective at preventing diseaseMedications for depression have severe side effectsI am likely to fully vaccinate my children/I have fully vaccinated my childrenComplete next section only if you interviewed a VPD-subjectCircle an answer: much more opposed, slightly more opposed, no effect, slightly more in favor, much more in favorHow did hearing about the subject’s physical suffering affect your opinion of vaccines?How did hearing about how the disease limited normal activity affect your opinion of vaccines?How did hearing about how the disease affected the subjects’ interactions with other people affect your opinion of vaccines?How did hearing about how the disease affected the subject’s family, friends, or loved ones affect your opinion of vaccines?How did hearing about the disease’s financial impact on the subject affect your opinion of vaccines?Rank the following:(a)Physical suffering(b)Limitation of activities(c)Interactions with other people(d)Effect on family, friends, or loved ones(e)Financial impactPlease explain briefly, what effect, if any, the project had on your attitude towards vaccination and why it has that effect.If this interview did NOT affect your attitude towards vaccination, why not?

### 2.3. Analyses

Changes between groups’ pre- and post-intervention VASs were assessed with factorial ANOVAs. Individual group changes over time were assessed by paired sample *t*-tests, and differences between two groups at specific time points were assessed by independent sample *t*-tests. Bonferroni corrections were applied to any multiple comparisons to account for alpha inflation. Standard deviations were reported for statistics less than 5 points. All other statistics reported 95% confidence intervals (CI). All analyses were performed using SPSS Statistics 25 (IBM). Figures were generated in Prism 8 (GraphPad) and tables were generated in Excel 2016 (Microsoft).

## 3. Results

### 3.1. Overview and Pre-Interview Intervention Vaccine Attitudes

A total of 574 students volunteered to take the pre-interview survey during the Winter 2018 semester. Based on their pre-intervention vaccination attitude scores (VASs), students were designated either pro-vaccine (PV) (87%) or vaccine-hesitant (VH) (13%) and assigned to the control group (autoimmune survey, *n* = 286) or the intervention group (vaccine-preventable disease (VPD) survey, *n* = 288). Of the students, 74% (*n* = 425) completed all requisite parts of the study (pre-interview survey, community/family interview, post-interview survey) and were included in the final analysis (Figure 1). The VH group was defined as VAS < 20 and the PV group was defined as VAS ≥ 20 based on the pre-interview survey responses. There were no statistically significant differences in the sociodemographic characteristics among the classes nor did the course they were enrolled in significantly affect the assignment to VH and PV groups (Table 1). Course year explains the age difference between the courses: Bio 100 is a general education course for first year students, MMBio 240 is a second-year major-specific course, and MMBio 261 is a second to third year major-specific course. Furthermore, sex, race, and age were not significantly correlated with pre-intervention vaccine attitudes (Table 2). Student willingness to vaccinate current/future children was significantly different between VH and PV groups (scale of 1–5 from strongly disagree to strongly agree) with means of 3.84 and 4.92, respectively (independent *t*-test CI 95% 0.814–1.355; *p* < 0.001).

### 3.2. Interview Intervention Improves Student Vaccine Attitude Scores

Vaccine attitudes improved when the participants gained a personal understanding of how vaccine-preventable diseases affect individuals and communities. Vaccine-hesitant students enrolled in MMBio 240 (no vaccine curriculum) who were part of the intervention group (*n* = 19) showed a significant increase in VAS; average VAS shifted from 17.58 ± 0.84 to 20.53 ± 0.94 (paired *t*-test CI difference (diff) 95% 4.077–0.817; *p* < 0.001), an average increase of 2.95 ± 2.34 points (Figure 2). Of these students, 68% (*n* = 13) had sufficient increases in their VASs to move from the vaccine-hesitant group to the pro-vaccine group. Conversely, vaccine-hesitant students who were part of the control group (*n* = 22) had no significant increase in VAS (paired *t*-test CI diff 95% 1.856–0.038; *p* = 0.059) which shifted only 17.27 ± 0.87 to 18.18 ± 1.31, an average increase of 1 ± 2.05 point (*p* = 0.059). Only 27% (*n* = 6) of students in the control group increased their scores sufficiently to move from the vaccine-hesitant group to the pro-vaccine group. Post-intervention VASs are significantly different between control and intervention VH groups (independent *t*-test CI diff 95% 4.066–0.623; *p* = 0.009), whereas post-intervention VASs are still significantly different between VH and PV students in the intervention group (independent *t*-test CI diff 95% 3.702–1.733; *p* < 0.001). α = 0.0125.

### 3.3. Vaccine Education Likely Improves Student Vaccine Attitudes

Intensive vaccine education may be even more effective at improving vaccine attitudes than interviewing individuals who have had a VPD. All vaccine-hesitant students (*n* = 5) enrolled in MMBio 261 (intensive immune, VPD, and vaccine education) significantly increased their VASs by 7.00 ± 1.41 points on average regardless of survey intervention (*p* < 0.001), (pre-control group MMBio 261 VH mean 16.50, CI 95% 14.41–18.59; post-control group VH mean 23.500, CI 95% 12.616–25.384; pre-intervention group VH mean 14.000, CI 95% 12.29–15.71; post-intervention group VH mean 21.00, CI 95% 19.46–22.54) (Figure 3a). For all VH students, including intervention and control groups, the pre-intervention VAS mean was 15.00 ± 2.06 and the post-intervention VAS mean was 22.00 ± 2.23, decidedly in the pro-vaccine range. Four out of five VH students increased their VASs sufficiently to move from the vaccine-hesitant category to the pro-vaccine category with an average increase of 7.50 ± 1.00 points. The final student, who participated in the intervention group, increased their VAS from 13 to 18 points, an increase of 5 points. There is a significant difference between pre- and post-intervention group MMBio 261 VH students (*p* = 0.026, *n* = 3 paired *t*-test CI diff 95% 11.97–2.03). Statistics cannot be run across time between pre- and post-control group VH students since there are a low number of respondents (*n* = 2). Although these results are promising, large in magnitude, and statistically significant, they are based on a small number of vaccine-hesitant students in the class (*n* = 5). Furthermore, students in MMBio 261 are majoring in a life sciences degree and may be more prone towards persuasion by scientific reasoning than the non-major students in the general education Bio 100 course.

To highlight the need for tailored and intensive vaccine education, vaccine-hesitant students in Bio 100 had a non-significant yet distinct upward trend over time regardless of survey intervention (Figure 3b). Overall, VASs do significantly change across time (*p* = 0.036) and vaccine attitude/survey groups (*p* < 0.001). All students in Bio 100 received brief instruction on how vaccines work, the rarity of vaccine side effects, the benefits of herd immunity to society, and no specific conversation about VPDs. The average VH student increased 1.9 ± 2.37 points between pre- and post-intervention VASs. There is no significant difference after the survey intervention between post-control group and post-intervention group VH students (independent *t*-test CI diff 95% 3.47–7.47; *p* = 0.42), or between post-control group PV and post-control group VH students (independent *t*-test CI diff 95% 8.32–1.57; *p* = 0.136). There is a significant difference after survey treatment between post-intervention group PV and post-intervention group VH students (independent *t*-test CI diff 95% 8.24–3.38; *p* < 0.001).

### 3.4. Vaccine-Hesitant Students’ VAS Change Dependent on Pre-Intervention VASs and Class

This intervention focuses on the vaccine attitudes and responses of vaccine-hesitant students to an interview intervention. To better understand what aspects of the interview intervention positively influenced VH students, we focused on analyzing the scores of VH students by comparing pre- and post-intervention VASs. Overall, most VH students (75%) have increased VASs, while 50% of all VH students advance to PV scores by the end of the study (Table 3). This gain, however, depends on class or interview group as previously described. For example, interview group determines the fate of MMBio 240 VH students but not Bio 100 VH students (Table 3). Class enrollment predicts pre- to post-intervention VAS changes (Figure 4a). MMBio 261 students have the greatest increase (7.50 ± 1.00), while students in Bio 100 and MMBio 240 had similar gains (3.40 ± 1.50 and 3.50 ± 2.00 points, respectively).

Yet, once students are broken into groups based on pre-intervention VASs, it becomes clear that not all VH students are alike (Figure 4b, Table 4). Students with the lowest pre-intervention VASs (11–15 points) are unlikely to become PV (*n* = 2.18%) and only gain an average of 2.91 ± 2.74 points (*p* < 0.001). This average is clearly defined by survey groups: intervention group students gain an average of 4.67 ± 2.65 points, whereas control group students gain an average of 0.8 ± 1.92 points. Students in this low score category who gained 5+ points (*n* = 4) were all part of the intervention group. This suggests that the most vaccine-hesitant students are swayed by VPD interviews. Students with middle VH pre-intervention VASs (16 or 17 points) gain an average of 4.00 ± 3.07 (*p* = 0.0095) and are more likely to become PV (*n* = 9, 60%). Overwhelmingly, students in this middle category who gained 5+ points were either in MMBio 261 (*n* = 3) or had conducted a VPD survey in Bio 100 or MMBio 240 (*n* = 4). Two students in this category were not in MMBio 261 and conducted autoimmune surveys, thus their reasons for change are not predictable. The final group of VH students with the highest pre-intervention VAS (18 or 19 points) gained the least, an average of 1.27 ± 2.02 points (*p* = 0.0018, *n* = 17.57%). As these students are near the highest range already, it is not surprising that no students gained more than 5 points as a 6 point gain places them at the top of the VAS range.

### 3.5. Vaccine-Hesitant Student Post-Intervention VAS Increase Correlated with Perceived Physical Suffering and Physical Limitations

In the post-survey interview, students in the intervention group were asked to assess how much each of the following characteristics affected their opinion of vaccines: physical suffering, limited normal activity, limited interaction with others, impact on family and friends, and financial costs (Methods, Box 3). These attributes were assessed from “strongly more opposed to vaccination” to “strongly more in favor of vaccination” and assigned the values of 1–5 points. VH students’ post-intervention VASs are significantly and moderately correlated with physical suffering (4.08 ± 0.845, r^2^ = 0.405, *p* = 0.04) (Figure 5a) and limitation on normal activities (3.88 ± 0.653, r^2^ = 0.518, *p* = 0.007) (Figure 5b). VH students with a positive pre- to post-intervention VAS change agree or strongly agree that physical suffering is of major importance; although the amount change in VAS compared to strength of agreement is not significant, there is a visible upward trend (one-way ANOVA, *p* = 0.3089) (Figure 5c). Even more strikingly, VH students with the greatest VAS change (6–9 points, *n* = 5) strongly agree that normal activity limitations affect their vaccine opinions (one-way ANOVA, *p* = 0.0206) (Figure 5d). This suggests that VH students are more influenced by stories from VPD victims that include physical suffering and activity limitation.

#### 3.5.1. Interview Examples Correspond to Student Perceptions of Physical Suffering and Physical Limitations

Examples of interview responses for physical suffering and physical limitations from students with the greatest VAS change (6–9 points) suggest that extreme cases enhance student response. One student interviewed a member of their church congregation who had shingles: “The pain was so bad that she ended up at a pain management clinic where they did steroid shots into her spine. The pain meds didn’t even touch [reduce] her pain, even the heavy ones. For months she couldn’t leave the house.” This interview led the student to explain (Methods, Box 3, question 25) that “The project showed how the lack of vaccination is essentially accepting the pain and suffering that comes with disease.” Another student interviewed his or her grandmother about tuberculosis: “Before getting diagnosed and during the time that she was treated, she could work her eight-hour temple shift and then she would go straight to bed after getting home. After a couple of hours nap, she would get up for a short time to get small tasks done before retiring to bed for the night.” This student summarized the interview experience as “I dislike the idea of physical suffering so hearing about someone getting a disease made the idea of getting a disease if I don’t get vaccinated seem more real.” These students both became PV with VAS increases of 7 and 6 points, respectively.

In keeping with this idea, many VH students with smaller VAS gains generally reported less serious physical suffering and physical limitations from the people they interviewed. A student who gained 4 points and interviewed a shingles patient wrote: “She considered her case very minor and she did not suffer physically much. She had some difficulty sleeping for a couple of weeks. She was a stay-at-home wife at that time, so she wasn’t missing work [or] school.” Similarly, a student who gained 3 points and interviewed a German measles case remarked, “Mother developed typical rash for about 3 days with high fever and remained bed bound. She is a school teacher and didn’t work for a few days.” While some VH students who gained low to middle VAS points had extreme examples, overall the tone was more moderate than the students who gained the greatest VAS points.

#### 3.5.2. Student Ranking of Influential Factors Does Not Match Actual Impact

We sought to confirm this finding by examining the ranking data (Methods, Box 3). We asked students to rank which factor (physical suffering, limitation of activities, interactions with other people, effects on family/friends, and financial impact) they perceived to have the greatest impact on the interviewee. Interestingly, 62% of all students ranked physical suffering as having the most impact, and 54% of students ranked financial costs as having the least impact. However, neither of these factors was significant for the whole population. There were no clear ranking distinctions for limitation of activities, and interactions with other people and family/friends. These findings are replicated for VH students. Therefore, in many cases, physical suffering of the interviewee is represented as the greatest symptom reported by the interviewer, but does not surpass other factors in influencing post-intervention VASs in general, at least as perceived by the student’s ranked understanding. This may be because what was most important to the interviewee does not necessarily directly correlate with the interviewers “take away”. For example, while students perceived that the interviewee might have emphasized characteristics other than physical suffering in their interview, the students themselves perceived pain as an important concept. This may be important to consider while designing interventions.

## 4. Discussion

In this study, we succeeded in improving student vaccine attitudes through either (1) having students interview individuals who had experienced a VPD or (2) providing intensive vaccine- and VPD-related course material. Combining intervention styles allowed us to assess the strength of each intervention. VPD interviews (intervention group) were most successful at swaying student vaccine attitudes when the coursework did not discuss vaccines or if the interviews had strong themes of physical suffering and limitations. The majority of students in the intervention group who became pro-vaccine and the resulting increase in vaccine attitude scores mirrored those achieved through intensive education (MMBio 261). Thus, encouraging students to conduct VPD interviews may be an easy and effective intervention when the course has little to do with VPDs or lacks vaccine-related content.

In courses that do address vaccines, it may be advantageous to first rigorously introduce students to VPD consequences before addressing, lightly, vaccine safety and societal implications. While Bio 100 introduces vaccines through a homework assignment that seeks to correct misconceptions about vaccine safety and societal implications, MMBio 261 begins with rigorous weeks-long sections on immunity and VPDs but only briefly discusses vaccine safety and herd immunity. This may explain why Bio 100 VH students did not have significantly increased post-intervention VASs, only a suggestive upward trend, whereas MMBio 261 VH students had significantly improved post-intervention VASs. The comparison between Bio 100 and MMBio 261 students mirrors earlier research that discussing VPD ramifications has a greater impact on combating vaccine hesitancy than correcting flawed assumptions or asserting an absence of risk about vaccines [11,12,13,14,15].

This study does have limitations. We did not examine whether an interview-based intervention would be successful in a non-science course. Any biological instruction discussing vaccines might provide some boost to vaccine attitudes. Additionally, for logistical reasons, we did not assess whether the increase in VAS is meaningful by following whether students vaccinate their current and future children. Furthermore, this study focuses on college students and may not be expandable to the general population. Nonetheless, despite these limitations, interview-based interventions and intensive VPD-dependent vaccine education does significantly increase vaccine attitudes, in a population susceptible to anti-vaccine attitudes. Vaccine hesitancy is a complex, situation-dependent problem, and requires unique and tailored interventions. Interview-based interventions are easy to implement and can supplement courses or even community outreach programs seeking to address vaccine hesitancy. Predisposing students to think more favorably about vaccinations by openly discussing the consequences of vaccine-preventable diseases may improve their prospective individual and familial vaccine uptake. Future research should tease apart the contributions of science education and personal familiarity with VPDs towards improving vaccine attitudes in diverse populations.

## 5. Conclusions

There are two major conclusions of this work. First, an interview-based intervention, where students discuss vaccine-preventable diseases with people who have actually experienced these diseases, can significantly improve attitudes towards vaccination. Second, the subject matter used while teaching about vaccine-preventable diseases matters. In the class with extensive discussion of the diseases themselves, there was a strong increase in vaccine attitudes among vaccine-hesitant students, while this effect was not seen in the class that discussed mostly vaccine safety. Taken together, these findings indicate that increasing familiarity with vaccine-preventable diseases leads to improved attitudes towards vaccination. This should help to create solutions to the worldwide problem of vaccine hesitancy or denial, by indicating aspects of education that are important for affecting those attitudes.

## Figures and Tables

**Figure 1 vaccines-07-00039-f001:**
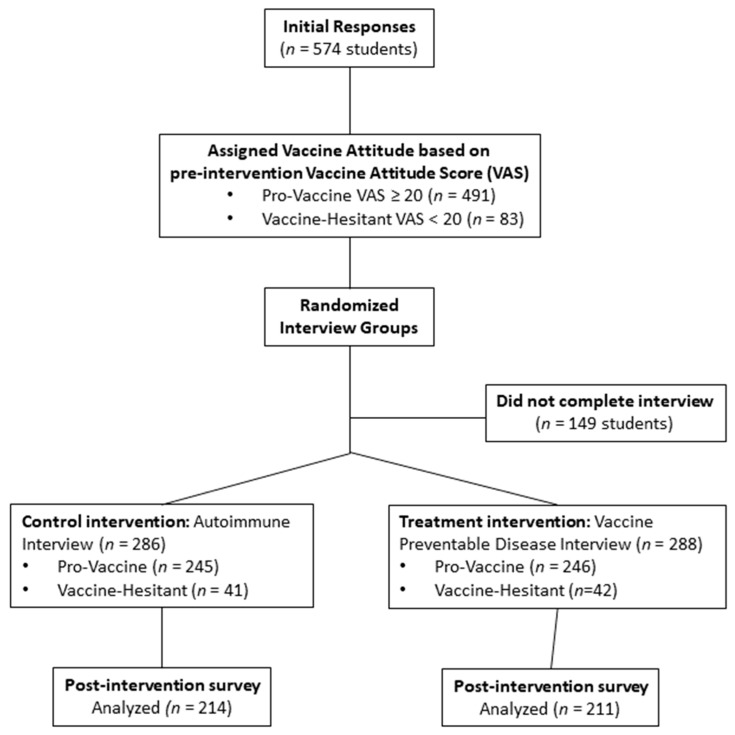
Participant flow through the randomized treatment.

**Figure 2 vaccines-07-00039-f002:**
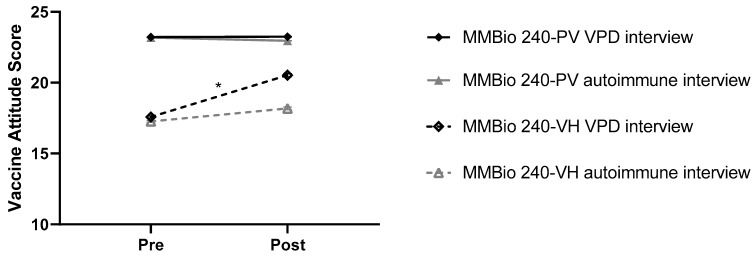
Vaccine-preventable disease interview significantly improves attitudes towards vaccines. Treatment makes a significant difference (interaction term *p* < 0.001) for vaccine-hesitant (VH) students in MMBio 240.

**Figure 3 vaccines-07-00039-f003:**
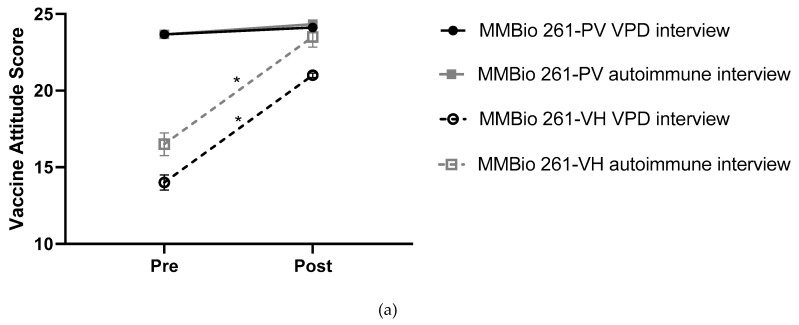
Education can significantly increase vaccine attitude. (**a**) Vaccine attitude scores (VASs) of MMBio 261 vaccine-hesitant students significantly increased regardless of survey intervention (*p* < 0.001), Difference between pre-control group VH and pre-intervention group MMBio 261 VH students is not significant (CI diff 95% 4.72–9.71; *p* = 0.35). (**b**) While there is an upward VAS trend for all Bio 100 VH students, it is not significant, suggesting that education has more influence than intervention.

**Figure 4 vaccines-07-00039-f004:**
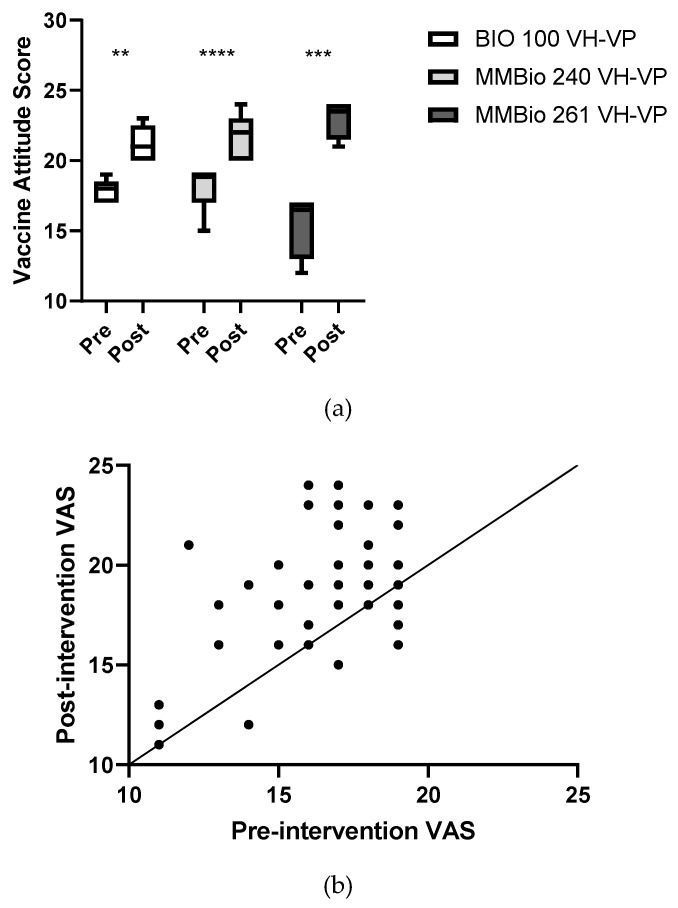
Vaccine-hesitant students make varying gains based on starting score and class attended. (**a**) VAS changes for VH students with PV post-intervention VASs. VH to PV students in MMBio 261 had an average VAS increase of 7.5 ± 1.0 points, whereas students in Bio 100 and MMBio 240 gained an average of 3.4 ± 1.5 and 3.5 ± 2.0 points, respectively. (**b**) VH students’ gains are determined by pre-intervention VASs. Plotting pre-intervention VASs against post-intervention VASs for VH students shows student responsiveness is dependent on pre-intervention VASs. The line indicates no change between pre-and post-intervention scores, so the farther away from the line the larger the change.

**Figure 5 vaccines-07-00039-f005:**
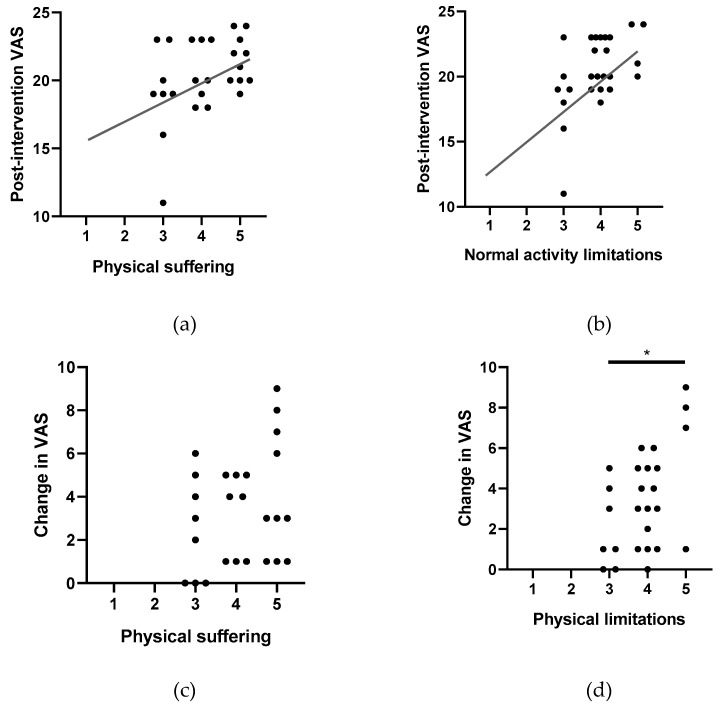
Post-intervention VAS and positive pre- to post-intervention VAS changes are influenced by (**a**,**c**) physical suffering and (**b**,**d**) physical activity limitations. (**a**) Post-intervention VAS is predicted by physical suffering (r^2^ = 0.405, *p* = 0.04) and (**b**) physical limitations (r^2^ = 0.518, *p* = 0.007). (**c**) While the student’s perception of physical suffering did not predict the amount of VAS change (*p* = 0.3089), (**d**) the student’s perception of normal activity limitations is significantly predicted (*p* = 0.0206). * *p* < 0.05.

**Table 1 vaccines-07-00039-t001:** Baseline characteristics of the participating classes (*n* = 425).

Class Demographics	Total % (*n*)	Bio 100 % (*n*)	MMBio 240 % (*n*)	MMBio 261 % (*n*)
Total	100% (425)	13% (56)	70% (298)	17% (71)
Gender				
Male	62% (263)	70% (39)	61% (182)	59% (42)
Female	38% (162)	30% (17)	39% (116)	41% (29)
Age	21.2 ± 0.21	19.7 ± 0.50	21.3 ± 0.24	22.1 ± 0.40
Pre-Intervention Vaccine Attitude Score				
Vaccine-Hesitant	13% (56)	18% (10)	14% (41)	7% (5)
Pro-Vaccine	87% (369)	82% (46)	86% (257)	93% (66)

There are no statistically significant differences among the classes for gender distribution, age, or Pro-Vaccine or Vaccine-Hesitant group assignment.

**Table 2 vaccines-07-00039-t002:** Baseline characteristics of vaccine-hesitant and pro-vaccine groups (*n* = 425).

Sociodemographic Characteristic	Total % (*n*)	Vaccine-Hesitant % (*n*)	Pro-Vaccine % (*n*)
Total	100% (425)	13% (56)	87% (369)
Gender			
Male	62% (263)	14% (36)	86% (227)
Female	38% (162)	12% (20)	88% (142)
Age	21.2 ± 0.21	21.0 ± 0.74	21.3 ± 0.21
Race/Ethnicity			
African American	1% (3)	-	100% (3)
Asian	3% (11)	27% (3)	73% (8)
Caucasian	87% (370)	13% (48)	87% (322)
Hispanic	3% (12)	17% (2)	83% (10)
Native American	0.2% (1)	100% (1)	-
Other	6% (26)	8% (2)	92% (24)

There were no statistically significant differences in ethnicity, gender, or age between vaccine opinion groups.

**Table 3 vaccines-07-00039-t003:** Survey intervention and education significantly improves vaccine-hesitant student VASs.

Change Post-Treatment	Overall VH % (*n*)	BIO 100 % (*n*)	MMBio 240 % (*n*)	MMBio 261 % (*n*)
Total	56	10	41	5
VAS Increased	75% (42)	80% (8)	71% (29)	100% (5)
VAS No Change	11% (6)	0% (0)	17% (7)	0% (0)
VAS Decreased	14% (8)	20% (2)	12% (5)	0% (0)
Pro-Vaccine VAS (20+ points)	50% (28)	50% (5)	46% (19)	80% (4)

Breakdown of all vaccine-hesitant (VH) post-intervention VASs regardless of survey treatment. All students who reached a VAS of 20+ were reassigned as pro-vaccine (PV). “VAS Increased” includes students who became PV.

**Table 4 vaccines-07-00039-t004:** Numerical breakdown of all VH students by pre-intervention VAS.

Pre-Intervention VAS	11–15 Points	16 or 17 Points	18 or 19 Points
Total	11	15	30
Post-intervention VAS			
VH	82% (9)	40% (6)	43% (13)
VP	18% (2)	60% (9)	57% (17)
Change (average)	2.91 ± 2.74	4.00 ± 3.07	1.27 ± 2.02
VAS Decreased	9% (1)	7% (1)	20% (6)
Age (average)	20.9 ± 0.9	21.0 ± 0.2	21.1 ± 1.3

All VH students broken down by pre-intervention VAS. Age is not significantly different between the groups.

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
