# Peer review of "Combating Vaccine Hesitancy with Vaccine-Preventable Disease Familiarization: An Interview and Curriculum Intervention for College Students"

_vaccines, 2019, doi:10.3390/vaccines7020039_

Round 1

Reviewer 1 Report

Dear Authors

The Article "Coming Vaccine Hesitancy with Vaccine Preventable Disease Familiarization: An Interview and Curriculum Intervention" is basically a well-written article and understandable. I simply suggest a couple of modifications that I think could improve the readability of the work.

In the results section I would suggest using Control Group and Intervened Group as names than interview and autoimmune interview. They are very similar and could cause confusion.

Regarding the analysis, it is correct but improvable. Linear regression models could have been used to explain the Score according to the rest of the variables. It is more complete than a simple ANOVA and some differences of means in paired groups.

I am not in favor of converting scores into qualitative variables for analysis, although I do support their interpretation. So, the last 3 graphs are confusing and could be better understood by means of a scatter plot.

The bibliography is correct as well as the limitations section.

There are few studies on interventions to improve predisposition to vaccination. Therefore, this study is new in terms of applied intervention. I think the authors should put more emphasis on the intervention and its consequences in vaccine coverage.

Author Response

Reply to Reviewer 1 Comments

Thank you for your comments which we feel have made the manuscript stronger. We have addressed the comments as follows.

1.  In the results section I would suggest using Control Group and Intervened Group as names than interview and autoimmune interview. They are very similar and could cause confusion.

The autoimmune survey group is now addressed as “control group” and VPD-survey group is now addressed as “intervention group”.  Corrections have been made throughout paper and figure legends.

2.  Regarding the analysis, it is correct but improvable. Linear regression models could have been used to explain the Score according to the rest of the variables. It is more complete than a simple ANOVA and some differences of means in paired groups.

Thank you for your suggestion.  We have run the linear regression and got similar results.  Following consultation with a statistics adviser, we determined that the original display of the results from the ANOVA is easier to understand and expresses the same information as the linear regression.

3.  I am not in favor of converting scores into qualitative variables for analysis, although I do support their interpretation. So, the last 3 graphs are confusing and could be better understood by means of a scatter plot.

Figure 4b has been redone as scatterplots and a 1 to 1 line added for clarity of image and to show the improvement in vaccine scores based on the pre-survey scores.

Figure 5a-d have been redone as scatterplots and trendlines were added to parts A and B. The discussion was revised in section 3.5 to discuss the ANOVA analysis as follows:

VH students post VAS scores are significantly and moderately correlated with physical suffering (4.08 ± 0.845, r2 = 0.405, p = 0.04) (Figure 5a)and limitation on normal activities (3.88 ± 0.653, r2 = 0.518, p = 0.007) (Figure 5b). VH students with positive pre- to post-intervention VAS change agree or strongly agree that physical suffering is of major importance; although the amount change in VAS compared to strength of agreement is not significant, there is a visible upward trend (one-way ANOVA, p = 0.3089) (Figure 5c). Even more strikingly, VH students with the greatest VAS change (6-9 pts, n = 5) strongly agree that normal activity limitations affect their vaccine opinions (one-way ANOVA, p = 0.0206) (Figure 5d). This suggests that VH students are more influenced by stories from VPD victims that include physical suffering and activity limitation.

 The figure references in the text have also been updated to reflect the new presentation of the data (lines 297, 298, 301, and 303).

4.  The bibliography is correct as well as the limitations section.

Thank you.

5.  There are few studies on interventions to improve predisposition to vaccination. Therefore, this study is new in terms of applied intervention. I think the authors should put more emphasis on the intervention and its consequences in vaccine coverage

We added the line: “Predisposing students to think more favorably about vaccinations by openly discussing the consequences of vaccine preventable diseases may improve their prospective individual and familial vaccine uptake.” (lines 349-351) to address this comment

Sincerely,

Brian Poole

Reviewer 2 Report

This study has several strengths: the researchers recruited participants from 3 classes of varying degrees familiarity vaccination, biology, and immunology to conduct interventions. The two types of interventions were adequately designed and the pre-post analysis were conducted correctly. It's biggest strength is that the study design supported the researchers' hypothesis that having students interview those with VPD encourages themselves to switch from VH to PV.

Some areas require revision: first, how was the pre-post test questionnaire designed was not adequately explained. Second, the figures are too wordy, try to incorporate the description in the results section. Third, the introduction can be strengthened by explaining why was interviewing people with VPD part of the design, any theories to support the design? 

Author Response

Reviewer 2

Thank you for your time in reviewing our manuscript. We feel that your comments have made the manuscript stronger. We have modified the manuscript as follows in response to your comments:

1. First, how was the pre-post test questionnaire designed was not adequately explained.

-We have inserted the following into the methods section when describing the design of the pre-intervention questionnaire:

Each question concerning vaccine attitude was chosen to cover a specific aspect of vaccine hesitancy. Pre-intervention Vaccine Attitude Scores (VAS) were tallied from questions 1, 4, 9, 11, and 13 (Box 1, bolded for clarity). Question 1 is a test of general attitude towards vaccines. Question 4 addresses vaccine side effects: relevant since many vaccine-hesitant individuals are afraid of these side effects. Question 9 is about the common belief that vaccines cause autism, a major concern for many vaccine hesitant people. Question 11 gives an opportunity for the participants to opine on the positive aspects of vaccination in terms of how well they work. Question 13 is a question about how the vaccine attitude would affect action, and give it a more real-world rather than theoretical effect. To avoid answer bias, students were not informed that the study was about vaccination opinions and additional questions about autoimmune diseases and depression were included in the survey. Scores from questions 4 and 9 were reverse coded to account for the negative nature of the question.  These questions were written in a negative way to avoid biasing the study by presenting vaccines in only a positive light in the questions.

-We also inserted the following to explain the post-intervention questionnaire:

These questions (14-18 in the Post-Interview Survey) were written to identify the aspects of the interview that had the most significant impact on vaccine attitudes. They provided both an opportunity to rank several factors and the ability to explain in their own words how the various aspects of the interview affected them.

We then assigned students this post-intervention VAS and assessed for changes in overall vaccine scores between the pre-and post-intervention surveys.

2. Second, the figures are too wordy, try to incorporate the description in the results section.

Table 1: shortened description to “There are no statistically significant differences among the classes’ for gender distribution, age or Pro-Vaccine or Vaccine Hesitant group assignment” and moved the remaining description to section 3.1 “Course year explains the age difference between the courses: Bio 100 is a general education course for first year students, MMBio 240 is a second year major-specific course and MMBio 261 is a second to third year major-specific course.”

Table 2: shortened description to “There were no statistically significant differences in ethnicity, gender, or age between vaccine opinion groups” and moved “VH group were defined as VAS < 20 and PV group were defined as VAS ≥ 20 based on the pre-interview survey responses.” to section 3.1.

Figure 2: shortened description to “Vaccine-preventable disease interview significantly improves attitudes towards vaccines. Treatment makes a significant difference (interaction term p < 0.001) for VH students in MMBio 240.” and interspersed the original statistic comments in the first paragraph of section 3.2. 

Figure 3: Shortened description to: “Education can significantly increase vaccine attitude. (a) VAS scores of MMBio 261 vaccine hesitant students significantly increased regardless of survey intervention (p <0.001), Difference between pre-control group VH and pre-intervention group MMBio 261 VH is not significant (CI diff 95% -4.72-9.71; p = 0.35). (b) While there is an upward VAS trend for all Bio 100 VH students, it is not significant, suggesting that education has more influence than intervention.” And enhanced the second paragraph of section 3.2 with the stats from the original figure legend.

Figure 4: no changes to 4a.  4b plot was redone per Reviewer 1’s request and the new description is shorter: “(b) VH students gains are determined by pre-VAS.  Plotting pre-VAS against post-VAS for VH students shows student responsiveness is dependent on pre-VAS score.”

Table 4: shortened to “All VH students broken down by pre-intervention VAS. Age is not significantly different between the groups.”

Figure 5: Graphs completely redone in response to Reviewer 1’s request, so the new legend was kept short.

3. Third, the introduction can be strengthened by explaining why was interviewing people with VPD
part of the design, any theories to support the design?

We inserted the following sentences to address this comment:

Additionally, the vaccine hesitant movement uses emotionally charged stories with dire long-term consequences to effectively convey anti-vaccine ideology.  Combating this rhetoric with a similarly emotional appeal may be an effective preventative strategy [18].  

Sincerely,

Brian Poole, PhD.